# Clinical Outcomes of Radiation Therapy for Angiosarcoma of the Scalp and Face: A Multi-Institutional Observational Study

**DOI:** 10.3390/cancers15143696

**Published:** 2023-07-20

**Authors:** Masanari Niwa, Natsuo Tomita, Taiki Takaoka, Hirota Takano, Chiyoko Makita, Masayuki Matsuo, Sou Adachi, Yukihiko Oshima, Shintaro Yamamoto, Mayu Kuno, Akifumi Miyakawa, Dai Okazaki, Akira Torii, Nozomi Kita, Seiya Takano, Motoki Nakamura, Hiroshi Kato, Akimichi Morita, Akio Hiwatashi

**Affiliations:** 1Department of Radiology, Nagoya City University Graduate School of Medical Sciences, Nagoya 467-8601, Japan; 2Department of Radiation Oncology, Gifu University Hospital, Gifu 500-1194, Japan; 3Department of Radiology, Aichi Medical University Hospital, Nagakute 480-1195, Japan; 4Department of Radiology, Japan Community Health Care Organization Chukyo Hospital, Nagoya 457-8510, Japan; 5Department of Radiation Oncology, Ichinomiya Municipal Hospital, Ichinomiya 491-8558, Japan; 6Department of Radiation Oncology, National Hospital Organization Nagoya Medical Center, Nagoya 460-0001, Japan; 7Department of Geriatric and Environmental Dermatology, Nagoya City University Graduate School of Medical Sciences, Nagoya 467-8601, Japan

**Keywords:** angiosarcoma, radiation therapy, chemotherapy, surgery, treatment outcome

## Abstract

**Simple Summary:**

Angiosarcoma of the scalp and face (ASF) is a rare, aggressive tumor often treated with multimodal therapy, including radiation therapy (RT). This study analyzed RT outcomes and prognostic factors in 68 non-metastatic ASF patients. Median radiation dose was 66 Gy in 33 fractions (interquartile range 60–70 Gy in 28–35 fractions). Local control (LC), progression-free survival (PFS), and overall survival (OS) rates were assessed. Higher LC rates were associated with an equivalent dose in a 2 Gy fraction (EQD_2_) >66 Gy. Combining chemotherapy or surgery with RT improved PFS rates. No factors affected OS. Late grade 3+ toxicities occurred in 1% of patients, including one with a grade 4 skin ulcer. These findings suggest that higher EQD_2_ (>66 Gy) and combination therapies enhance LC and PFS in ASF. Further studies are needed to optimize treatment strategies for this rare malignancy, particularly in elderly patients.

**Abstract:**

Angiosarcoma of the scalp and face (ASF) is a rare, aggressive tumor often treated with multimodal therapy, including radiation therapy (RT). This study assessed RT outcomes for ASF and identified prognostic factors. Data from 68 non-metastatic ASF patients undergoing RT with or without other therapies were analyzed. Median radiation dose was 66 Gy in 33 fractions (interquartile range (IQR) 60–70 Gy in 28–35 fractions). Local control (LC), progression-free survival (PFS), and overall survival (OS) rates were calculated using Kaplan–Meier analysis. Multivariate analyses and adverse event evaluation were conducted. Median patient age was 75 years (IQR 71–80 years), with a median follow-up of 17 months (IQR 11–42 months). One-/three-year LC rates were 57/37%, PFS rates were 44/22%, and OS rates were 81/44%. Multivariate analyses showed that an equivalent dose in a 2 Gy fraction (EQD_2_) >66 Gy correlated with improved LC (HR 2.35, 95% CI 1.03–5.32, *p* = 0.041). Combining chemotherapy (HR 2.43, 95% CI 1.08–5.46, *p* = 0.032) or surgery (HR 2.41, 95% CI 1.03–5.59, *p* = 0.041) improved PFS. No factors influenced OS. Late grade 3+ toxicities occurred in 1%, with one patient developing a grade 4 skin ulcer. These findings suggest that EQD_2_ > 66 Gy and combining chemotherapy or surgery can enhance LC or PFS in ASF. Further prospective studies are needed to determine the optimal treatment strategy for this rare malignancy, particularly in elderly patients.

## 1. Introduction

Angiosarcoma is a rare and highly malignant tumor of vascular endothelial cells. It accounts for approximately 2% of all soft tissue sarcomas and most frequently occurs in the scalp and face of elderly males [1,2]. Angiosarcoma of the scalp and face (ASF) initially presents as multiple red or dark purple plaques, followed by tumor growth, infiltration, edema, ulceration, and bleeding. This extremely aggressive tumor spreads widely through the skin, frequently recurs locally, and rapidly metastasizes early. A high incidence of pulmonary metastasis often leads to the occurrence of hemopneumothorax, resulting in a poor prognosis for patients [3,4,5]. The reason for a poor prognosis is not only frequent recurrence at the local site and/or the lungs, but also ASF being standard in the very elderly.

Surgery has been considered the standard treatment for ASF. However, since many lesions are multifocal or ill-defined, local recurrence is common even after extensive surgical resection [5,6]. Therefore, ASF is currently treated with multimodal therapy, including surgery, radiation therapy (RT), and chemotherapy [3,7,8,9,10,11,12]. On the other hand, statistical analyses in most studies on treating ASF have been limited by the small number of patients [7,8,9,10]. Furthermore, although several advanced RT techniques, such as intensity-modulated radiation therapy (IMRT), have improved clinical outcomes for many cancers [13,14], few studies on ASF have investigated the effects of these new techniques on outcomes. Therefore, the present study examined the clinical outcomes of non-metastatic ASF treated with RT with or without other anticancer therapies and attempted to identify potential prognostic factors for an optimal therapeutic approach.

## 2. Materials and Methods

### 2.1. Patient Selection and Pretreatment Evaluation

This was a retrospective, observational study approved by the Institutional Review Board of our institution (Approval number: 60-22-0043) and other participating hospitals. We reviewed ASF patients treated with RT with or without other anticancer therapies at six institutions between 1999 and 2021. Eligibility criteria were defined as follows: (1) histologically confirmed angiosarcoma; (2) primary lesion at the scalp and/or face; and (3) no distant metastasis. Seventy-one patients with non-metastatic ASF were treated with RT between 1999 and 2001 at six hospitals. Three patients were excluded from this analysis due to the complete lack of follow-up data. Therefore, 68 patients were reviewed for this study. The present study was performed by the ethical standards laid down in the 1964 Declaration of Helsinki and its later amendments. Since this was a retrospective observational study, informed consent was obtained in the form of opt-out on the website.

Clinical staging was based on the clinical presentation and a computed tomography (CT) scan of the head, neck, chest, and upper abdomen. Magnetic resonance imaging (MRI) and ^18^F-fluoro-deoxyglucose positron emission tomography (PET)/CT were performed on 10 (15%) and 35 patients (51%), respectively.

### 2.2. RT

Except for 36 patients treated with electron beams alone, 32 underwent CT-based radiation planning using thermoformed contention masks to maintain a reproducible position throughout the treatment. The gross tumor volume (GTV) was defined by the clinical presentation and CT and/or MRI and/or PET/CT. Before performing CT scans, dermatologists and radiation oncologists attached markers on the skin around the gross lesion to facilitate recognition of the GTV. In local irradiation, the clinical target volume (CTV) ranged at least 3 cm radially along the scalp from the edge of the GTV. In whole-scalp irradiation, CTV was defined as the whole scalp including the GTV. A supplementary circumferential margin of 3–5 mm was applied around the CTV to offset positional and physical uncertainties to create the planning target volume (PTV). Six patients (9%) received unilateral neck RT, four of whom had cervical lymph node metastases. Electron beams or three-dimensional conformal RT (3DCRT) were used for 41 patients (60%) and intensity-modulated RT (IMRT) for 27 (40%). In IMRT plans, conformal isodose distributions were created with the sparing of nearby normal tissues, such as the lens, eye, brain, oral cavity, and parotid glands. The median RT dose and number of fractions were 66 Gy in 33 fractions (interquartile range (IQR) 60–70 Gy in 28–35 fractions). The median fractionated dose was 2.0 Gy (IQR 2.0–2.0 Gy). The equivalent dose in a 2 Gy fraction (EQD_2_) with α/β = 10 was 66.0 Gy (IQR 60–70 Gy).

### 2.3. Combination Therapy

Six patients (9%) received RT alone. Seven patients (10%) underwent RT and surgery. Forty-two patients (62%) received combination therapy of taxane-based chemotherapy with RT. Twenty patients (29%) underwent surgery with pre- or postoperative RT. Thirteen patients (19%) received trimodality therapy with surgery, chemotherapy, and RT.

Among 20 patients (29%) who underwent surgery, surgical margins at the primary tumor were positive in 8 (12%) and negative in 12 (18%). Two and eighteen patients underwent preoperative and postoperative RT, respectively. Three patients (4%) received interleukin-2 immunotherapy.

### 2.4. Follow-Up Evaluation and Statistical Analysis

After RT, a clinical examination was performed at 2- or 3-month intervals. A head, neck, chest, and upper abdominal CT scan was performed at least every 4 months. PET/CT was conducted whenever recurrence was suspected. Tumor responses were assessed based on a complete response (CR), partial response (PR), progressive disease (PD), and stable disease (SD) with Response Evaluation Criteria in Solid Tumors (RECIST) [15].

Local control (LC), progression-free survival (PFS), and overall survival (OS) rates were calculated using the Kaplan–Meier method. LC was defined as the time from the start date of RT to local relapse, which was considered to be regrowth within the primary tumor or a new lesion on the scalp and face. PFS was defined as the time from the start date of RT to any recurrence or death as events and was censored at the last date without events. OS was defined as the time from the start date of RT to the last follow-up or death from any cause. The prognostic significance of variables was examined using a univariate analysis with the log-rank test and a multivariate analysis with a Cox proportional hazard model. In consideration of the number of patients and survival events, major potential factors were selected from previous multivariable analyses [3,4,10,16,17,18,19,20,21,22,23]. All statistical analyses were performed using EZR, which is a graphical user interface for R (The R Foundation for Statistical Computing, Vienna, Austria) [24]. RT-related morbidities were graded according to the National Cancer Institute Common Terminology Criteria for Adverse Events (NCI-CTCAE, version 5.0).

## 3. Results

### 3.1. Patient Characteristics

Table 1 shows patient and treatment characteristics. The median age of patients was 75 years (IQR 71–80 years). All patients were diagnosed with angiosarcoma by a histopathological examination of biopsy specimens. Five patients (7%) had cervical lymph node metastases, while none had distant metastases. Twenty-five patients (37%) had a tumor >5 cm and nineteen (28%) had multiple tumors. Twenty-eight patients (41%) had bleeding from their tumors.

### 3.2. Tumor Responses

Tumor responses were unknown in two patients (3%) with missing data and eighteen (26%) who underwent postoperative RT. In another 48 cases (71%), tumor responses were evaluated at the point of maximum tumor reduction within 1–3 months after RT. Of these patients, 9 (18%), 29 (58%), 8 (16%), and 2 (4%) patients achieved CR, PR, SD, and PD, respectively. The response rate, including CR and PR, was 76%.

### 3.3. Survival

The median follow-up period was 17 months (IQR 11–42 months) for all patients and 19 months (IQR 10–54 months) for living patients. One-/three-year LC, PFS, and OS rates were 57% (95% confidence interval [95% CI], 43–68%)/37% (95% CI, 23–52%), 44% (95% CI, 31–55%)/22% (95% CI, 12–34%), and 81% (95% CI, 68–89%)/44% (95% CI, 30–57%), respectively.

Thirty-seven patients (54%) died. Thirty-one patients (46%) died of the progression of ASF, while six (9%) died of other causes. Thirty-five patients (51%) developed local relapse. The median period to local relapse was 7 months (IQR 5–13 months). Eleven patients (16%) developed cervical lymph node metastases. The median period to the appearance of cervical lymph node metastases was 7 months after RT (IQR 6–12 months). Thirty-nine patients (57%) developed distant metastases. The median period to the appearance of distant metastases was 9 months (IQR 5–15 months). The first sites of distant metastases included the lung (28 cases), bone (2 cases), liver (2 cases), brain (1 case), mediastinal lymph node (1 case), lung and liver (2 cases), lung and bone (1 case), lung and kidney (1 case), and lung and abdominal cavity (1 case).

### 3.4. Univariate and Multivariate Analyses

Table 2 shows the log-rank test results for LC, PFS, and OS. In the log-rank test for LC, sex and the use of IMRT had significant effects. The LC rate was higher in female patients than male patients (1-year rates for females vs. male, 77% vs. 49%, *p* = 0.028, Figure 1A). The LC rate was also higher in patients treated with IMRT than those treated with 3DCRT and/or electron beams (1-year rates for IMRT vs. non-IMRT, 69% vs. 49%, *p* = 0.044, Figure 1B). The irradiation field and EQD_2_ also affected LC. The LC rate was slightly higher in the whole-scalp group than in the local irradiation group (1-year rates for whole scalp vs. local irradiation, 71% vs. 51%, *p* = 0.071, Figure 1C). The LC rate was also slightly higher in the EQD_2_ > 66 Gy group than in the EQD_2_ ≤ 66 Gy group (1-year rates for >66 Gy vs. ≤66 Gy, 70% vs. 42%, *p* = 0.056, Figure 1D).

In the log-rank test for PFS, only chemotherapy had a significant effect. The PFS rate was significantly higher in patients treated with combination chemotherapy than in those not receiving chemotherapy (1-year rates for chemotherapy use vs. non-use, 46% vs. 37%, *p* = 0.035, Figure 1E). In the log-rank test for OS, the OS rate was significantly higher in patients <75 years than in those ≥75 years (1-year rates for <75 vs. ≥75, 96% vs. 67%, *p* = 0.005, Figure 1F). The OS rate was also slightly higher in female patients than male patients (1-year rates for females vs. males, 88% vs. 78%, *p* = 0.066).

Table 3 shows the results of univariate COX regression analysis for LC, PFS, and OS. Male patients were associated with a lower LC rate (hazard ratio [HR] 2.57, 95% CI 1.06–6.25, *p* = 0.037). The combination of chemotherapy was associated with a higher PFS rate (hazard ratio [HR] 1.98, 95% CI 1.02–3.86, *p* = 0.045). The OS rate was higher in patients <75 years than in those ≥75 years (hazard ratio [HR] 2.58, 95% CI 1.23–5.11, *p* = 0.007).

Based on the number of patients and survival events, five major potential factors were selected for the multivariable analysis: tumor size (≤5 cm vs. >5 cm), the number of tumors (solitary vs. multiple), chemotherapy (use vs. non-use), surgery (use vs. non-use), and EQD_2_ (>66 Gy vs. ≤66 Gy). Table 4 summarizes the results from multivariable analyses of LC, PFS, and OS. EQD_2_ > 66 Gy was associated with a higher LC rate (hazard ratio [HR] 2.35, 95% CI 1.03–5.53, *p* = 0.041). The combination of chemotherapy or surgery was associated with a higher PFS rate (chemotherapy: HR 2.43, 95% CI 1.08–5.46, *p* = 0.032; surgery: HR 2.41, 95% CI 1.03–5.59, *p* = 0.041). No factors associated with OS were identified in the multivariable analysis. The use of chemotherapy was associated with a slightly higher OS rate (*p* = 0.092).

### 3.5. Adverse Events

Two patients developed grade 2 or higher late adverse events. One patient had grade 2 dysgeusia and dry mouth, while the other had grade 2 dysgeusia, dry mouth, and a grade 4 skin ulcer. Grade 4 skin ulcers are those with necrosis extending to muscle, bone, or supporting tissues. The cumulative incidence of late ≥ grade 2 and late ≥ grade 3 toxicities was 3 and 1%, respectively. The 70-year-old male patient who developed an ulcer on his scalp after surgery received postoperative RT of 66 Gy in 30 fractions (2.2 Gy per fraction) with paclitaxel. Since the ulcer worsened and exposed the skull, reconstructive surgery was performed 51 months after RT.

## 4. Discussion

Angiosarcomas are rare, aggressive tumors from vascular structures, are often located in the head and neck, and have a poor prognosis, particularly when they occur on the scalp and face [2,3,25]. Previous studies conducted retrospective analyses to develop an optimal therapeutic approach for ASF; however, statistical analyses in most studies were limited by the small number of patients examined. To the best of our knowledge, the present study is the most extensive series of ASF examined to date. Angiosarcoma reportedly has a male-to-female ratio of approximately 2:1 and occurs in the elderly [26,27]. In the present study, the male-to-female ratio was 74/26% and the median age of patients was 75 years. Patient characteristics in this study were similar to those in previous studies [26,27]. The results obtained herein confirmed the poor clinical outcomes of ASF. Sasaki et al. reported the outcomes of 30 patients with angiosarcoma treated with RT, including 6 with angiosarcoma other than ASF [28]. One-year LC and distant metastasis-free survival rates and the 13-year OS rate were 57, 37, and 20%, respectively. Ogawa et al. also described the outcomes of 48 patients with ASF after RT [4]. Two-year LC and PFS and one-year OS rates were 46, 10.7, and 22%, respectively. In the present study, 1-/3-year LC, PFS, and OS rates were 57/37%, 44/22%, and 81/44%, respectively. LC and PFS rates in the present study were consistent with those in previous studies. The higher OS rate in the present study than in other studies may be attributed to a good performance status (PS) of less than 1 in most patients (90%). For example, 20 out of 48 patients (42%) had PS 2 or 3 in the study by Ogawa et al.

Our multivariate analysis confirmed that the combination of chemotherapy and surgery significantly increased PFS, which was consistent with previous findings [4,18,22,29,30]. The combination of RT and surgery has been identified as the optimal treatment to decrease local recurrence [21,22,31,32,33]. Ihara et al. reported a 3-year PFS rate of 100% with the combination of RT and surgery, suggesting that surgery is an integral component of the treatment of ASF [29]. However, since most patients with ASF are very elderly, radical surgery is not often feasible. In addition, complete resection is technically difficult if the tumor is too large or there are too many tumors. Although the size and number of tumors were not independent prognostic factors in the present study, they have been shown to affect outcomes [4,7,16,23,28,34]. Therefore, the efficacy of chemotherapy becomes important in the treatment of ASF patients. Previous studies demonstrated that paclitaxel-based regimens improved survival [9,35,36] because taxane-based chemotherapy exerts radiosensitizing effects [30]. Fujisawa et al. found significant improvements in OS in 16 patients treated with taxane-based chemotherapy [9]. Schlemmer et al. [31] also noted slight improvements in OS and PFS in 32 patients with advanced angiosarcoma treated with paclitaxel. Further clinical studies are needed to confirm the effects of chemotherapy on survival and to assess its role in multimodal therapy for ASF. Recombinant IL-2 immunotherapy was previously used to enhance the effects of RT [32] but is not commonly performed any longer due to the lack of evidence.

Previous studies identified age, PS, tumor size, the number of tumors, radiation dose, bleeding from tumors, and the use of multimodal therapy as prognostic factors for ASF [1,3,4,10,16,17,18,19,20,21,22,23,28,29,31,35]. In our univariate analysis, sex and RT methods, including IMRT, the irradiation field, and radiation dose, appeared to have affected LC. Buehler et al. showed that male sex was a predictive factor for a poor prognosis in postoperative ASF [36], which is consistent with the present results. IMRT has become more common in recent years because it provides an excellent homogeneous dose distribution over the whole scalp while reducing the irradiated dose and volume to organs at risk (OAR), such as the brain and parotid gland [37,38]. Previous studies suggested that total scalp irradiation was a more effective treatment than local irradiation [37,39,40]. Mizuno et al. [41] compared PTV coverage and doses to OAR among helical tomotherapy, volumetric-modulated arc therapy, and intensity-modulated proton therapy (IMPT) and showed that all three techniques achieved sufficient coverage and satisfactory homogeneity for PTV. The present study appears to be the first to examine the clinical efficacy of IMRT for ASF patients. The results obtained suggested that EQD_2_ > 66 Gy was associated with more favorable LC. Suzuki et al. [35] reported that local recurrence correlated with the development of distant metastasis. Therefore, a dose escalation to the gross lesion may be important to improve LC and prevent distant metastasis.

Patients with ASF frequently develop pulmonary metastases with a hemopneumothorax, which is often fatal [3,4,5,21,32]. In the present study, 39 out of 68 patients developed distant metastases after RT, 31 of whom initially had lung metastases. Therefore, systemic treatment may be necessary to prevent the distant metastasis of ASF. In the multivariate analyses, the combination of chemotherapy was associated with better PFS (HR 2.43, 95% CI 1.08–5.46, *p* = 0.032).

In previous studies, response rates were 50–89% [11,31,42,43]. The response rate in the present study was 76% (9 with CR and 29 with PR). Fata et al. [43] treated nine ASF patients with multimodality treatment using paclitaxel and reported a response rate of 89% (four with CR and four with PR). To improve response rates, multimodality treatment with paclitaxel may be useful for ASF.

We encountered one case of a grade 4 skin ulcer as a late morbidity. This patient developed an ulcer on his scalp after surgery and was treated with postoperative RT at 66 Gy in 30 fractions before the ulcer healed. In addition, concurrent paclitaxel was combined with RT. Multiple factors, such as the premature start of postoperative RT, the concurrent combination of chemotherapy, and an increased dose per fraction may have caused this severe toxicity.

The present study has several limitations inherent to its retrospective design. The relatively short median follow-up of 17 months may have been insufficient to accurately detect local tumor progression and late adverse events. Furthermore, only four patients had lymph node metastases, which were not properly investigated. Moreover, some patients were not followed up at the predetermined intervals, and data were missing at some time points. In addition, since this was a multi-institutional study, the assessment of tumor responses requiring inspection and adverse events may not have been consistent. Given these limitations, the present results need to be interpreted with caution.

## 5. Conclusions

The present study is currently one of the largest series of ASF treated with RT. The poor outcomes observed were similar to those in previous studies. The results herein suggest that EQD_2_ > 66 Gy and the combination of chemotherapy or surgery improved LC or PFS, respectively. Although a significant effect was not found, the use of IMRT and total scalp irradiation may be effective for LC. A prospective study is needed to establish an optimal treatment strategy for this rare malignancy, which is common in the very elderly.

## Figures and Tables

**Figure 1 cancers-15-03696-f001:**
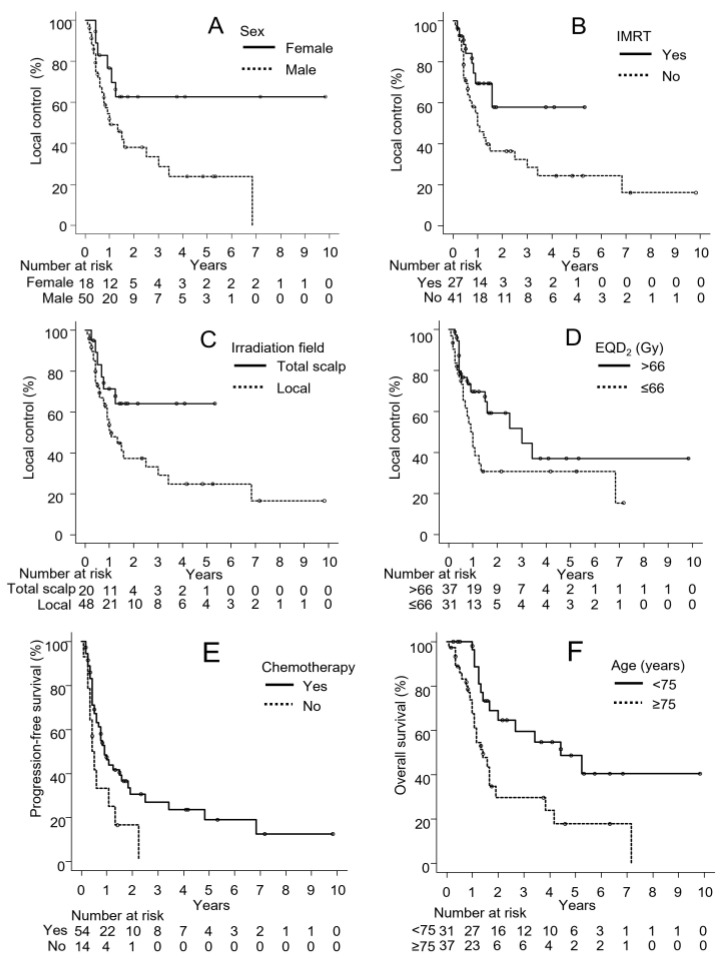
(**A**) Differences in local control by sex, (**B**) the use of intensity-modulated radiation therapy (IMRT), (**C**) the irradiation field, and (**D**) the equivalent dose in a 2 Gy fraction (EQD_2_). (**E**) Differences in progression-free survival by the use of chemotherapy. (**F**) Differences in overall survival by age.

**Table 1 cancers-15-03696-t001:** Patient and treatment characteristics.

Characteristics	*n* = 68
Sex	
Male/female	50 (74%)/18 (26%)
Age (years)	75 (IQR 71–80)
<75/≥75	31 (46%)/37 (54%)
Tumor size (cm)	
≤5/>5	43 (63%)/25 (37%)
Number of tumors	
Solitary/multiple/missing	45 (66%)/19 (28%)/4 (6%)
Node metastases	
No/yes	64 (94%)/4 (6%)
Performance Status	
0/1/2	40 (59%)/21 (31%)/7 (10%)
Bleeding from tumors	
No/yes	40 (59%)/28 (41%)
Treatment methods	
RT	6 (9%)
RT + surgery	7 (10%)
RT + chemotherapy ± immunotherapy	42 (62%)
RT + surgery + chemotherapy ^a^ ± immunotherapy ^b^	13 (19%)
Irradiation field	
Local/Total scalp	48 (71%)/20 (29%)
Irradiation methods	
IMRT/3DCRT or electron beams	27 (40%)/41 (60%)
Total dose (Gy)	66 (IQR 60–70)
EQD_2_ (Gy)	66 (IQR 60–70)

RT = radiation therapy, IMRT = intensity-modulated radiation therapy, 3DCRT = three-dimensional conformal radiation therapy, EQD_2_ = equivalent dose in a 2 Gy fraction with α/β  =  10 Gy, IQR = interquartile range. ^a^: The most common schedule is weekly dosing for 3 weeks followed by a 1-week break. ^b^: Intratumoral or arterial injection of interleukin-2 at the same time as radiation therapy.

**Table 2 cancers-15-03696-t002:** Univariate analysis of potential prognostic variables for survival.

Characteristics	Variables	*n*	Local Control		Progression-Free Survival		Overall Survival	
			1-Year Rate (%)	*p*-Value	1-Year Rate (%)	*p*-Value	1-Year Rate (%)	*p*-Value
Sex	Female	18	77	0.028	54	0.082	88	0.066
	Male	50	49		40		78	
Age (years)	<75	31	69	0.15	53	0.18	96	0.005
	≥75	37	46		37		67	
Tumor size (cm)	≤5	43	60	0.93	50	0.92	77	0.91
	>5	25	53		35		87	
Number of tumors	Solitary	45	57	0.57	40	0.28	78	0.39
	Multiple	19	59		50		82	
Nodal disease	No	64	58	0.68	44	0.82	81	0.76
	Yes	4	50		50		75	
Chemotherapy	Yes	54	60	0.14	46	0.035	81	0.086
	No	14	43		33		77	
Surgery	Yes	20	61	0.86	55	0.52	83	0.89
	No	48	55		39		80	
IMRT	Yes	27	69	0.044	41	0.94	76	0.69
	No	41	49		45		84	
Irradiation field	Total scalp	20	71	0.071	36	0.84	80	0.65
	Local	48	51		47		81	
EQD_2_ (Gy)	>66	37	70	0.056	47	0.41	76	0.57
	≤66	31	42		39		86	
Bleeding from tumors	No	40	57	0.61	48	0.29	87	0.14
	Yes	28	57		38		72	

IMRT = intensity-modulated radiation therapy, EQD_2_ = equivalent dose in a 2 Gy fraction with α/β  =  10 Gy.

**Table 3 cancers-15-03696-t003:** Univariate Cox regression analysis.

	Local Control			Progression-Free Survival			Overall Survival		
Variable	HR	95% CI	*p*-Value	HR	95% CI	*p*-Value	HR	95% CI	*p*-Value
Sex									
0: Female	2.57	1.06–6.25	0.037	1.82	0.90–3.66	0.10	2.21	0.92–5.31	0.08
1: Male									
Age									
0: <75	1.63	0.83–3.21	0.16	1.46	0.82–2.58	0.20	2.58	1.23–5.11	0.007
1: ≥75									
Tumor size (cm)									
0: ≤5	0.97	0.48–1.94	0.93	0.97	0.54–1.74	0.92	0.96	0.49–1.88	0.91
1: >5									
Number of tumors									
0: solitary	0.80	0.38–1.70	0.57	0.71	0.37–1.35	0.29	0.73	0.35–1.51	0.40
1: multiple									
Nodal disease									
0: no	0.74	0.18–3.11	0.68	0.89	0.32–2.48	0.82	1.20	0.37–3.93	0.76
1: yes									
Chemotherapy									
0: yes	1.81	0.81–4.03	0.15	1.98	1.02–3.86	0.045	1.87	0.90–3.89	0.09
1: no									
Surgery									
0: yes	1.07	0.51–2.23	0.86	1.23	0.65–2.32	0.53	1.05	0.52–2.14	0.89
1: no									
IMRT									
0: yes	2.20	0.99–4.86	0.052	0.98	0.54–1.77	0.94	0.87	0.44–1.74	0.70
1: no									
Irradiation field									
0: total scalp	2.18	0.90–5.28	0.08	1.07	0.56–2.02	0.84	1.19	0.57–2.46	0.65
1: local									
EQD_2_ (Gy)									
0: >66	1.89	0.96–3.71	0.06	1.26	0.72–2.22	0.42	1.20	0.63–2.30	0.58
1: ≤66									
Bleeding from tumors									
0: no	0.83	0.41–1.71	0.62	1.353	0.76–2.42	0.31	1.66	0.84–3.30	0.15
1: yes									

HR = hazard ratio, 95% CI = 95% confidence interval, IMRT = intensity-modulated radiation therapy, EQD_2_ = equivalent dose in a 2 Gy fraction with α/β  =  10 Gy.

**Table 4 cancers-15-03696-t004:** Multivariate analysis of major potential prognostic variables for survival.

	Local Control			Progression-Free Survival			Overall Survival		
Variable	HR	95% CI	*p*-Value	HR	95% CI	*p*-Value	HR	95% CI	*p*-Value
Tumor size (cm)									
0: ≤5	1.10	0.50–2.43	0.81	1.00	0.53–1.91	0.99	1.11	0.54–2.30	0.77
1: >5									
Number of tumors									
0: solitary	0.56	0.23–1.40	0.22	0.56	0.26–1.19	0.13	0.66	0.28–1.56	0.34
1: multiple									
Chemotherapy									
0: yes	1.77	0.64–4.89	0.27	2.43	1.08–5.46	0.032	2.11	0.88–5.05	0.092
1: no									
Surgery									
0: yes	2.00	0.72–5.53	0.18	2.41	1.03–5.59	0.041	1.85	0.73–4.67	0.19
1: no									
EQD_2_ (Gy)									
0: >66	2.35	1.03–5.32	0.041	1.35	0.69–2.63	0.38	1.08	0.52–2.24	0.84
1: ≤66									

HR = hazard ratio, 95% CI = 95% confidence interval, EQD_2_ = equivalent dose in a 2 Gy fraction with α/β  =  10 Gy.

## Data Availability

The data supporting this study are not publicly available due to the privacy of research participants but are available from the corresponding author on reasonable request.

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
