# Peer review of "Clinical Outcomes of Radiation Therapy for Angiosarcoma of the Scalp and Face: A Multi-Institutional Observational Study"

_cancers, 2023, doi:10.3390/cancers15143696_

Round 1

Reviewer 1 Report

From a biostats and clinical epidemiology point of view, this research has been well planned and reported. I have only some minor comments for the Authors.

- line 24/35, please, add IQR to median value

- line 27 Combining chemotherapy "or" surgery improved PFS rates, maybe you mean "combining to RT"?

- everywhere, continuous covariates are to be described as median/IQR, without using range

- line 150, tumor responses, at what timing? at the end of RT?

- line 178, I do remember you the ideal events/covariate ratio for multivariate survival models is 8-10, do you believe that your models could be a bit overparametrized?

- line 178, I do recommend to estimate a full series of Cox PH univariate models, rather than to rely on log-rank results; then, to report these data together with those coming from the Cox PH multivariate models 

- table 1, some details about the most commonly used CT and IT schedules could be of help for the reader

- figure 1, very informative!

Author Response

We thank you for providing constructive comments regarding the improvement of the original manuscript. Here, we are sending our revised manuscript. All changes have been made in response to the suggestion, and itemized response to the individual comments are also described.

1) Reviewer #1 comment: line 24/35, please, add IQR to median value

Author response: Thank you for your comment. We have added IQR to the median value as suggested.

2) Reviewer #1 comment: line 27 Combining chemotherapy "or" surgery improved PFS rates, maybe you mean "combining to RT"?

Author response: Thank you for your comment. We have added the phrase "to RT" to clarify the intended meaning.

3) Reviewer #1 comment: everywhere, continuous covariates are to be described as median/IQR, without using range.

Author response: Thank you for your comment. We have described continuous covariates as median/IQR instead of using range.

4)  Reviewer #1 comment: line 150, tumor responses, at what timing? at the end of RT?

Author response: Thank you for your comment. We have provided additional details regarding the timing of tumor response assessment.

5) Reviewer #1 comment: line 178, I do remember you the ideal events/covariate ratio for multivariate survival models is 8-10, do you believe that your models could be a bit overparametrized?

Author response: Thank you for your comment. As you mentioned, the ideal events/covariate ratio for multivariate survival models is 8-10. In this study, 37 patients died. Thirty-one patients died of the progression of ASF, while 6 died of other causes. Thirty-five patients developed local relapse. Based on these event numbers, we determined that the appropriate number of covariates for our analysis was 5. Furthermore, we took into consideration previous literature with a smaller sample size, which typically showed 3-6 covariates. Considering these findings, we believe that our chosen number of covariates is appropriate for this study.

6) Reviewer #1 comment: line 178, I do recommend to estimate a full series of Cox PH univariate models, rather than to rely on log-rank results; then, to report these data together with those coming from the Cox PH multivariate models.

Author response: Thank you for your comment. We have estimated a full series of Cox PH univariate models. We have also added a mention of this in the manuscript and created a new Table 3. We decided to retain Table 2 as it was difficult to remove it while presenting the survival curves in Figure 1. We have made some changes to the order of "Yes" and "No" in Table 2 to align with Table 3.

7) Reviewer #1 comment: table 1, some details about the most commonly used CT and IT schedules could be of help for the reader

Author response: Thank you for your comment. We have added some details about the most commonly used CT and IT schedules to Table 1.

8) Reviewer #1 comment: figure 1, very informative!

Author response: Thank you for your positive feedback on Figure 1. We appreciate your recognition of its informative nature.

Thank you once again for the invaluable feedback provided by the reviewers. We are grateful for their time, expertise, and dedication in evaluating our work.

Reviewer 2 Report

The authors analyze a topic which is of interest – angiosarcoma of the scalp and face, a rare but aggressive tumor.

The presentation is clear, comprehensive and well documented.

The references are appropriate, up-to-date and contain 43 titles.

I found no self-citations.

The figure is appropriate and mandatory for sustaining the topic.

The 3 tables offer concentrated information on the topic.

I found no plagiarism.

The discussions and conclusions are coherent and connected to the content.

It is a retrospective study, fact underlined by the authors, but patient selection and pretreatment evaluation, therapy (RT, Combination therapy) , follow-up evaluation and statistical analysis, results (tumor responses, survival, adverse event) were thoroughly documented.

In my opinion the paper fits the journal and the language is correct and understandable.

A small correction: page 2 line 59 - The reason for a poor prognosis is not only frequent recurrence at the local site and/or the lungs, but also  ASP being standard in the very elderly. Instead of ASP it should be ASF.

I recommend the paper to be accepted.

Author Response

We would like to express our heartfelt gratitude to the reviewers for their valuable time, expertise, and thoughtful evaluation of our manuscript. We sincerely appreciate your assessment and positive feedback, which have greatly contributed to the improvement of our work.

1) Reviewer #2 comment: A small correction: page 2 line 59 - The reason for a poor prognosis is not only frequent recurrence at the local site and/or the lungs, but also ASP being standard in the very elderly. Instead of ASP it should be ASF.

Author response: Thank you for your comment. We have replaced "ASP" with "ASF" to accurately reflect the intended meaning.

Thank you once again for the invaluable feedback provided by the reviewers. We are grateful for their time, expertise, and dedication in evaluating our work.
